# Suppressed electronic contribution in thermal conductivity of Ge$_2$Sb$_2$Se$_4$Te

Kiumars Aryana[1], Yifei Zhang [2], John A. Tomko [1], Md Shafkat Bin Hoque [1], Eric R. Hoglund [3], David H. Olson[1], Joyeeta Nag[4], John C. Read[4], Carlos Ríos [5,6], Juejun Hu [2] & Patrick E. Hopkins [1,3,7✉]

Integrated nanophotonics is an emerging research direction that has attracted great interests for technologies ranging from classical to quantum computing. One of the key-components in the development of nanophotonic circuits is the phase-change unit that undergoes a solid-state phase transformation upon thermal excitation. The quaternary alloy, Ge$_2$Sb$_2$Se$_4$Te, is one of the most promising material candidates for application in photonic circuits due to its broadband transparency and large optical contrast in the infrared spectrum. Here, we investigate the thermal properties of Ge$_2$Sb$_2$Se$_4$Te and show that upon substituting tellurium with selenium, the thermal transport transitions from an electron dominated to a phonon dominated regime. By implementing an ultrafast mid-infrared pump-probe spectroscopy technique that allows for direct monitoring of electronic and vibrational energy carrier life-times in these materials, we find that this reduction in thermal conductivity is a result of a drastic change in electronic lifetimes of Ge$_2$Sb$_2$Se$_4$Te, leading to a transition from an electron-dominated to a phonon-dominated thermal transport mechanism upon selenium substitution. In addition to thermal conductivity measurements, we provide an extensive study on the thermophysical properties of Ge$_2$Sb$_2$Se$_4$Te thin films such as thermal boundary conductance, specific heat, and sound speed from room temperature to 400 °C across varying thicknesses.

[1] Department of Mechanical and Aerospace Engineering, University of Virginia, Charlottesville, VA 22904, USA. [2] Department of Materials Science and Engineering, Massachusetts Institute of Technology, Cambridge, MA, USA. [3] Department of Materials Science and Engineering, University of Virginia, Charlottesville, VA 22904, USA. [4] Western Digital Corporation, San Jose, CA 95119, USA. [5] Department of Materials Science and Engineering, University of Maryland, College Park, MD 20742, USA. [6] Institute for Research in Electronics and Applied Physics, University of Maryland, College Park, MD 20742, USA. [7] Department of Physics, University of Virginia, Charlottesville, VA 22904, USA. ✉email: phopkins@virginia.edu

Modern computing relies on the processing of information by constantly shuttling the data back and forth between the storage and the processing units[1]. This computing architecture, known as von Neumann, leads to huge traffic jams between the memory and processor, incurring considerable costs in terms of latency and energy[2,3]. With the growing demand for data-centric technologies such as artificial intelligence and machine learning, there is a global effort to find alternative computing paradigms to supersede the traditional von Neumann architecture[4–7]. Biologically-inspired neuromorphic computing is one of the more promising alternatives to transistor-based technologies that not only offers a significantly higher degree of connectivity between the memory nodes leading to faster computation and less power consumption, but also allows for simultaneous storage and processing of information within the memory cell[8,9]. Chalcogenide-based phase change materials (PCMs) have been an indispensable component in the development of this technology due to their large properties contrast at different phases[10]. With the emergence of these materials, several works have demonstrated that neuromorphic computing is possible both in the realm of nanoelectronics and nanophotonics[11,12].

In chalcogenide-based PCMs, thermal excitation can induce reversible solid-state phase transitions between amorphous and crystalline states[13,14]. This phase transition is non-volatile and leads to large contrasts in the electrical[15], optical[16], and thermal properties[17]. Germanium antimony telluride, Ge−Sb−Te (GST), is the most popular and the most studied chalcogenide-based PCMs due to its phase stability[14], large property contrast[18], and fast switching rate[19,20]. Although GST has been successfully implemented into a number of different applications from thermal camouflage[21] to reconfigurable metalenses[22], its properties are not optimized for photonic devices. For instance, in nanophotonic devices such as optical memories and reconfigurable meta-optics, both phases of GST suffer from large optical losses[16] which limits its implementation.

Recently, a class of phase change materials, $Ge_2Sb_2Se_4Te$ (GSST), has emerged which offers superior properties in regards to photonic applications, such as broadband transparency for the wavelengths in the range of 1–18.5 μm, significant refractive index contrast ($\Delta n$) between the phases with low optical loss contrast ($\Delta k$) leading to a large figure of merit ($\Delta n/\Delta k$), and improved thermal stability[23–25]. Despite the growing interests for the integration of GSST into optical and photonic devices, such as reconfigurable metasurfaces, its thermal properties remain unknown.

Understanding the thermal transport properties of PCMs is of critical importance to design and modeling of active photonic devices based on PCMs. Switching PCM in photonic devices is customarily performed using micro-heaters made of metals[24,26], doped Si[27–29], transparent conductive oxides[30,31], and graphene[32,33]. In all cases, the heater design must meet several requirements to enable reversible PCM switching, including sufficient heating temperature to trigger phase transition, rapid quench rate to facilitate re-amorphization, and maximal temperature uniformity to ensure device longevity[34]. To fulfill these requirements, extensive thermal modeling of the micro-heaters must be carried out. Thermal properties of the heater materials, in particular the PCMs, are therefore essential for their applications in photonics.

For this, we investigate the thermal properties of GSST across phase transition for thicknesses ranging from 20 to 220 nm. Specifically, we report on the thermal conductivity, longitudinal sound speed, and volumetric heat capacity of GSST in amorphous and crystalline phases (see Table 1). We find that although GSST is a close cousin to $Ge_2Sb_2Te_5$ composition, their respective heat transport mechanisms in the crystalline phase are fundamentally different. In particular, we show that upon substituting Te with Se in GST, the thermal transport is dominated by vibrational carriers rather than electrons across different phases. These results are understood from a series of ultrafast time-domain spectroscopy methods that allow for direct investigation of electrons and phonons scattering rates in these materials. Specifically, we experimentally observe a drastic reduction in the electronic lifetimes of GSST in comparison to its GST counterpart. We attribute this increased electron scattering rate in GSST to the presence of an additional atomic species that introduces both intrinsic mass scattering as well as a change in the local bonding environment of the material system.

## Results

The GSST films were prepared using thermal evaporation from a single $Ge_2Sb_2Se_4Te$ source. The bulk starting material of $Ge_2Sb_2Se_4Te$ was synthesized using a standard melt quench technique from high-purity (99.999%) raw elements. The film deposition was performed using a custom-designed system (PVD Products, Inc.) following previously established protocols[16]. Stoichiometries of the films were confirmed using wavelength-dispersive spectroscopy (WDS) on a JEOL JXA-8200 SuperProbe Electron Probe Microanalyzer (EPMA) to be within 2% (atomic fraction) deviation from target compositions. In order to confirm compositional homogeneity across the film thickness, we use energy-dispersive X-ray spectroscopy in a scanning transmission electron microscope (TEM) (see Supplementary Note 1).

To investigate the atomic structure of GSST thin-films before and after phase transformation, we perform selected area electron diffraction in a transmission electron microscope on amorphous and crystalline samples that are nominally 20 and 150 nm thick as shown in Fig. 1. Each of the four diffraction patterns shows a high-intensity, single crystalline, [110] zone-axis Si diffraction pattern from the oriented substrate. In addition to the Si substrate pattern, Fig. 1b and f show the diffraction patterns of the amorphous sample have diffuse rings which result from short- and medium-range order and a lack of long-range order. The lack of other diffraction peaks demonstrates that the samples are purely amorphous and lack any crystalline structure. Diffraction patterns from the 150 nm crystalline sample (Fig. 1d) exhibits many Bragg diffraction peaks, in addition to the single-crystalline substrate diffraction pattern, indicating the existence of many nanometer-sized crystalline grains. Diffuse rings from an amorphous GSST material are still present in this 150 nm selected area diffraction pattern (Fig. 1d), which could suggest incomplete crystallization or minor damage from sample preparation; similar observations are found in the 20 nm crystalline GSST film. A ~5% thickness reduction is measured upon crystallization in both 20 and 150 nm films indicating densification of GSST after the phase transformation, similar to previous observations in GST[17,35].

The thermophysical properties of these PCMs are measured using time-domain thermoreflectance (TDTR), an optical pump-probe thermometry technique that is capable of measuring thermal properties of thin films such as thermal conductivity, thermal boundary conductance, specific heat, and sound speed. The details regarding measurement technique and the thermal model that relates the experimental data to thermal properties of the films studied here are discussed in detail elsewhere[36–39].

To understand the thermal transport dynamics in our GSST films, we perform a suite of ultrafast experiments on various thicknesses and at different temperatures. First, we investigate the frequency-dependent thermoreflectance signal in our TDTR measurements; by varying the modulation frequency of which the thin film is heated, the thermal decay varies from an effusivity

| **Table 1 The thermal properties of GSST studied here in amorphous and crystalline phases for 150 nm GSST.** | | | | | |
|---|---|---|---|---|---|
| GSST phase | Thermal conductivity (W m$^{-1}$ K$^{-1}$) | Specific heat (MJ m$^{-3}$ K$^{-1}$) | Density (g cm$^{-3}$) | Sound speed (m s$^{-1}$) | GSST/Al$_2$O$_3$ TBC (MW m$^{-2}$ K$^{-1}$) |
| Amorphous | 0.20 ± 0.02 | 1.5 ± 0.1 | 5.27[32] | 2300 ± 100 | - |
| Crystalline | 0.48 ± 0.06 | 1.8 ± 0.1 | 5.53[a] | 2750 ± 150 | 20 |
| [a]Considering 5% densification upon amorphous-to-crystalline phase transformation. | | | | | |

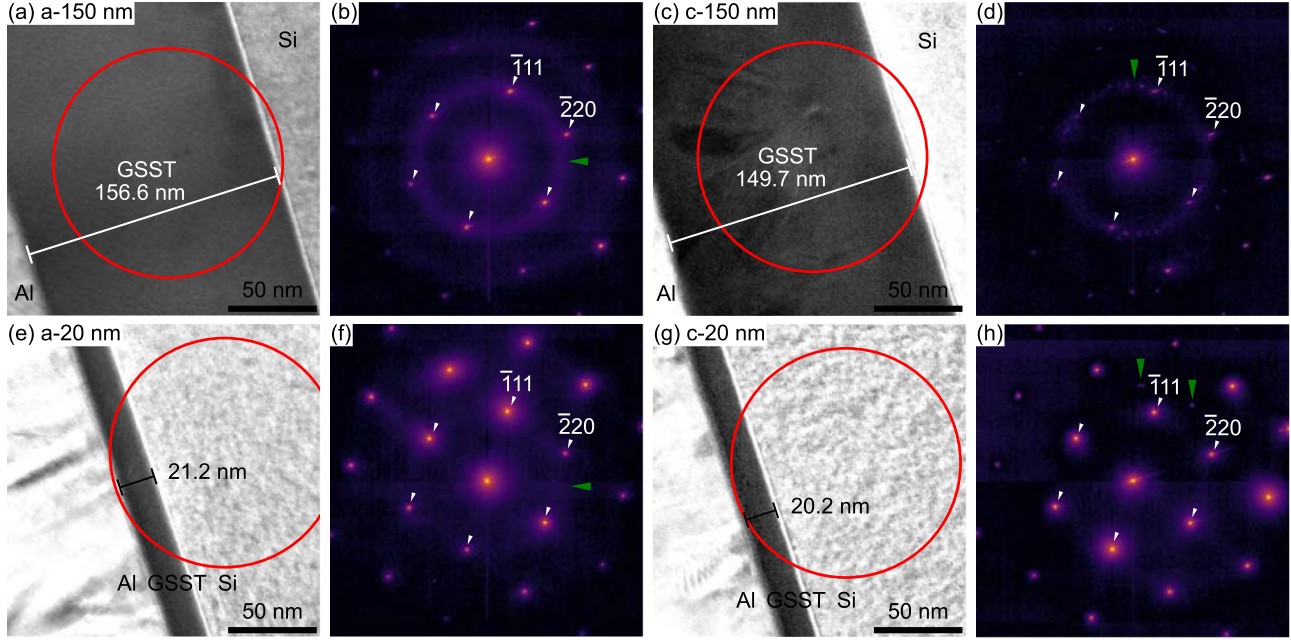

**Fig. 1 The structure and thickness variation in GSST upon phase transformation.** Bright-field TEM images and the selected area electron diffraction patterns for 150 nm (**a**, **b**, **c**, **d**) and 20 nm (**e**, **f**, **g**, **h**) films in amorphous and crystalline phases. Red circles indicate the selected area where diffraction patterns are acquired. In all cases, the selected area aperture includes the Si substrate. In diffraction patterns (**b**, **d**, **f**, **h**), white arrows indicate first-order reflections from the Si substrate and the green arrows point to the first-order GSST (**b**, **f**) amorphous ring, **d** polycrystalline ring, and isolated reflections.

regime to a purely diffusivity-dominated regime. In other words, these frequency-dependent TDTR experiments allow us to independently measure both the volumetric heat capacity and thermal conductivity of our GSST films without convolution from other thermophysical properties of the heterostructure. We choose modulation frequencies in the range of 0.4–8.4 MHz for this purpose and perform the experiment on the thickest (~220 nm) amorphous and crystalline (annealed at 300 °C) samples. This thickness is chosen because our measurements have the highest sensitivity to its thermal conductivity and volumetric heat capacity. Figure 2a shows the result of these heat capacity measurements where the intersection of our data at different modulation frequencies corresponds to the volumetric heat capacity[40]. Based on these data, the volumetric heat capacity for the amorphous and annealed cases are determined to be 1.5 ± 0.1 and 1.8 ± 0.1 MJ m$^{-3}$ K$^{-1}$, respectively. Phase transformation in all the samples was independently verified using micro-Raman measurements (Fig. 2b).

Second, we measure the longitudinal sound speed in GSST via picosecond acoustics, the details of which are given elsewhere[17,41,42]. In these measurements, the pump pulse launches strain waves on the surface of the sample that propagates through the underlying layers with the speed of sound. These strain waves, upon reaching an interface between the two materials, depending on the acoustic mismatch, are partially reflected and partially transmitted. The reflected portion of the waves travel all the way back to the surface and can be detected by our

probe beam with picosecond resolution. Figure 2c shows the corresponding peaks and troughs due to reflection from the Al/GSST and GSST/Si interfaces for 20 and 150 nm thick GSST. By measuring the time between these two peaks and the knowledge of film thicknesses, we can obtain the sound speed. Using this technique, we measure the sound speed in GSST for the two different phases at various thicknesses as shown in Fig. 2d. We observe that although the sound speed in amorphous GSST remains relatively constant across different thicknesses, the sound speed in the crystalline phase converges to that of the amorphous phase as the thickness of GSST film decreases. This behavior has also been observed in GST thin films[17]. Although identifying the underlying reasons behind this reduction is beyond the scope of this work, we hypothesize that it is due to partial crystallization of GSST near the interfaces for the thinner films.

For investigating the thermal conductivity, we begin our experiments by measuring GSST at room temperature. For this, we deposit different thicknesses of GSST films with 7 nm Al$_2$O$_3$ coating to avoid reaction between GSST film and the metallic transducer. We then take samples with different thicknesses and anneal them at 300 °C to form a uniform crystalline phase. The thickness of GSST films studied here is selected in the range of 20–220 nm, which captures both of its thin-film and bulk-like thermal properties. We measure thermal resistance across transducer-substrate (Pt/Al$_2$O$_3$/GSST/Si) that incorporates the resistance due to all interfaces and interlayers[17]. By solely changing the thickness of GSST film, we can vary the relative

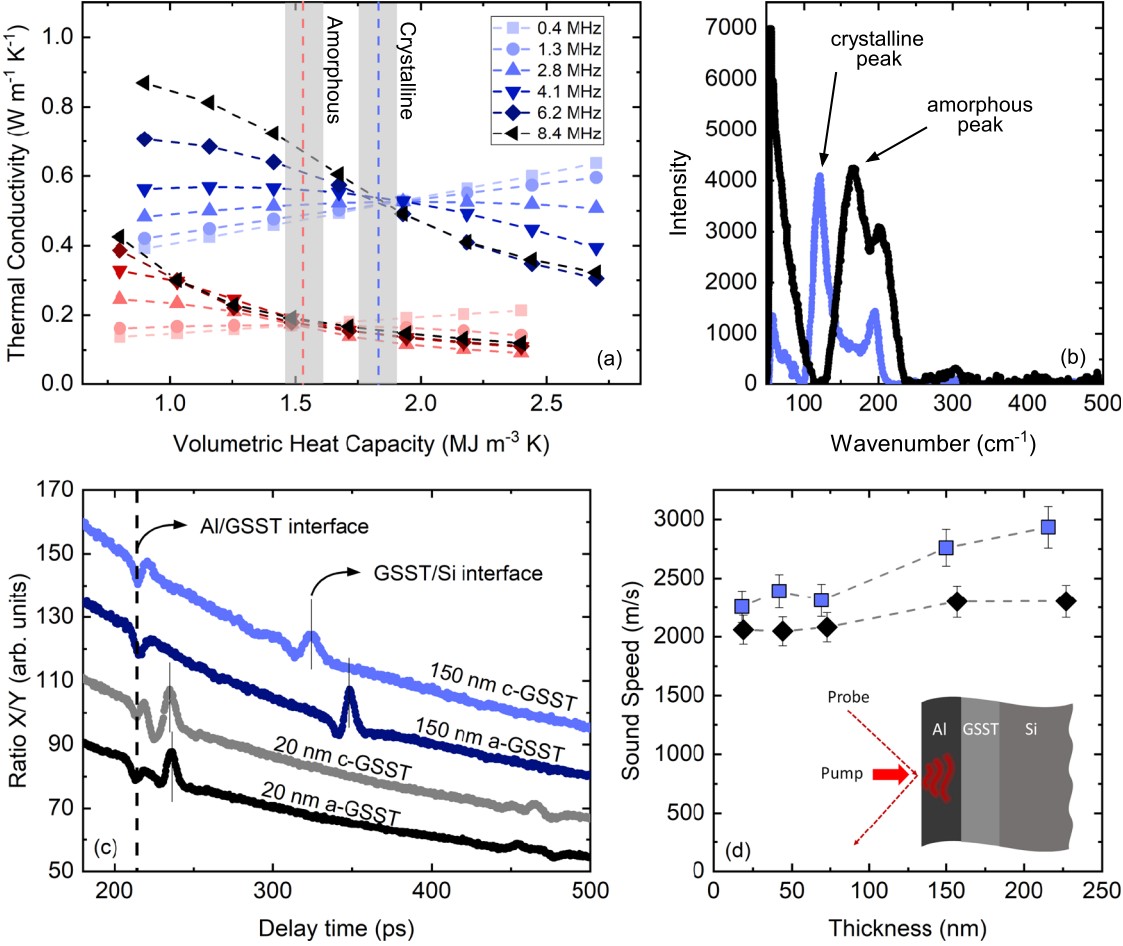

**Fig. 2 Thermophysical properties of GSST upon phase transformation. a** GSST volumetric heat capacity for amorphous and crystalline (annealed at 300 °C) phases measured with TDTR at different modulation frequencies. The gray area corresponds to the uncertainty of the measurements. **b** Raman spectra for GSST showing the amorphous and crystalline peaks, **c** picosecond acoustics measurements and the corresponding echoes from the interfaces, **d** sound speed for different thicknesses of GSST in amorphous (black diamonds) and crystalline (blue squares) phase. The uncertainty is calculated based on 10% variation in GSST film thickness.

contribution of GSST to the resistance of the stack and hence find its thermal conductivity independent of other extraneous resistances such as interfaces and $Al_2O_3$ layer (see Supplementary Note 2). Figure 3a shows the thermal resistance as a function of the GSST thickness where the inverse of the linear fit to the experimental data corresponds to the thermal conductivity of the GSST. According to this, the thermal conductivity of the GSST at room temperature for the as-deposited amorphous and 300 °C annealed crystalline phases are 0.20 and 0.48 W m$^{-1}$ K$^{-1}$, respectively.

In order to understand the thermal transport mechanisms of GSST across amorphous-to-crystalline phase transition, we measure the thermal conductivity of GSST at elevated temperatures using a resistive heating stage. For this, we perform the measurement on the thinnest (20 nm) and the thickest (220 nm) samples where we have variable sensitivities to the interfacial thermal resistance and thermal conductivity. For the case of 20 nm thick GSST, due to small resistance from the film itself, the resistance from the interfaces would dominate the thermal transport. Figure 3b shows the thermal conductivity of 20 nm thick GSST as a function of temperature. According to this figure, the thermal conductivity of 20 nm GSST does not show a noticeable change across its crystalline phase transition, or at temperatures up to 400 °C. Although this might lead one to conjecture that the phase transformation did not take place due to the reduced thickness of the film, we confirm that the

phase transformation does in-fact occur via both Raman and TEM measurements. This behavior is very similar to our previous work[17] where we demonstrated that the thermal conductivity of the 20 nm GST can be reduced by a factor of four compared to thick films (160 nm) due to the dominance of interfacial resistance. Similarly here, we attribute the temperature-independent thermal conductivity of the 20 nm GSST to the dominance of interfacial thermal resistance. With the knowledge of thermal conductivity, we estimate the thermal boundary conductance (TBC) between $Al_2O_3$ and crystalline GSST to be 20 MW m$^{-2}$ K$^{-1}$. Note, we do not report a TBC for the amorphous phase as our measurements do not have sufficient sensitivity to it[17,43].

For the case of 220 nm GSST, due to the large thermal resistance from the film itself, our TDTR measurements directly capture the intrinsic thermal conductivity of the GSST film with minimal influence from the resistances at interfaces. As shown in Fig. 3b, the thermal conductivity in the amorphous phase is approximately 0.18 W m$^{-1}$ K$^{-1}$, in strong agreement with the value we obtained from the linear fit to the thickness series in the previous section. For the 220 nm GSST, we observe that the thermal conductivity remains constant with increasing temperature up to the onset of crystallization and shows a modest increase upon phase transition at ~180 °C. This is consistent with previous measurements for the temperature at which the phase transformation occurs in GSST[16]. After phase transformation, the

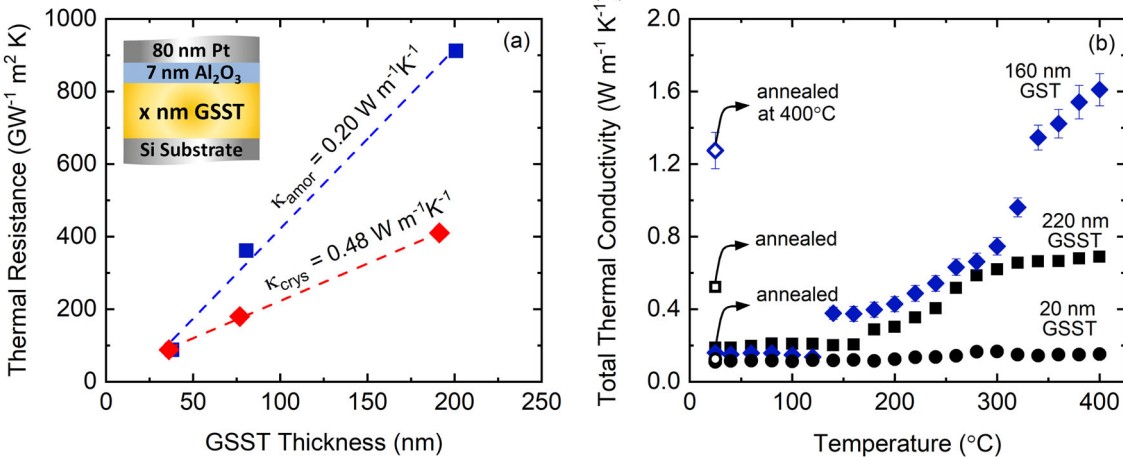

**Fig. 3 Measured thermal conductivity. a** GSST thermal resistance as a function of thickness in amorphous and crystalline (annealed at 300 °C) phases. The inverse of slope for the linear fit to the experimental data corresponds to the thermal conductivity in each phase. **b** Thermal conductivity of as-deposited GSST with thicknesses of 20 nm (solid circles) and 220 nm (solid squares) as a function of temperature. The thermal conductivity GST (solid diamonds) is shown for comparison.

thermal conductivity gradually increases with temperature up to ~280 ° C after which the thermal conductivity reaches a constant value of ~0.65 W m$^{-1}$ K$^{-1}$ up to 400 °C. For comparison, we plot the thermal conductivity data for 160 nm GST, which represents the intrinsic thermal conductivity of GST independent of interfaces similar to 220 nm GSST. By comparing the thermal conductivity of GST vs. GSST as a function of temperature, we observe that the thermal conductivity is significantly suppressed in the crystalline GSST by more than a factor of two. In order to investigate how thermal properties GST change upon Se substitution from a pure vibrational mode perspective, i.e. disregarding the contribution from electrons, both Se and Te are 2-coordinated elements, and therefore, Se substitutions should not alter the degree of connectivity and bonding network in the system, leaving minimal impact on the thermal conductivity from a topological perspective[44]. However, the insertion of Se as an additional element is expected to increase the scattering rate. At the same time, Se is a lighter element compared to Te and should lead to an increase in thermal conductivity. As a result of these competing events and to gain a deeper understanding of the dominant phonon scattering mechanisms in GSST, we turn to analytical models that provide an estimation for the thermal conductivity based on the kinetic theory of gases, generally referred to as the minimum limit to thermal conductivity[45]. According to Cahill and Pohl[46], the minimum scattering length of vibrational carriers, in this case, phonons, is on the order of one half of their wavelength:

$$\kappa_{\min,\mathbf{P}} = 1.21 \, k_{\mathrm{B}} \, n^{2/3} \, v_g, \tag{1}$$

where $k_{\mathrm{B}}$ is the Boltzmann constant, $v_g$ is the average sound speed of the material, and $n$ is the number density. We calculate the average sound speed with respect to the longitudinal ($v_{\mathrm{LA}}$) and transverse ($v_{\mathrm{TA}}$) modes as: $v_g^2 = \frac{1}{3}\left(2v_{\mathrm{TA}}^2 + v_{\mathrm{LA}}^2\right)$. Assuming density of 6.20 and 5.53 g cm$^{-3}$, and sound speed of 2800 and 2750 m s$^{-1}$ for crystalline phases of GST and GSST, respectively[17,32], we calculate the minimum limit to the thermal conductivity of the GST and GSST to be 0.36 and 0.38 W m$^{-1}$ K$^{-1}$, respectively. Based on this, from a pure phononic perspective, the thermal conductivity of GST with its higher mean atomic mass is almost similar to that of the GSST. However, from previous studies, we know that the thermal conductivity of GST in the hexagonal phase is primarily dominated by the electronic contribution due to metal-insulator transition[47–49]. As a result, we conjecture that the observed reduction in thermal

conductivity of GSST is related to the suppressed contribution of electrons rather than phonons. To investigate this, we estimate the electronic contribution in thermal conductivity by using the Wiedemann−Franz (W−F) formalism that related electrical resistivity to the thermal conductivity:

$$k_{\mathrm{e}} = LT/\rho \tag{2}$$

where $L$ is the Lorenz number $2.44 \times 10^{-8}$ W Ω K$^{-2}$, $T$ is the absolute temperature, and $\rho$ is the electrical resistivity. Based on previously measured electrical resistivity for GST and GSST as a function of temperature[16], we plot the electronic contribution in thermal conductivity in Fig. 4a. According to these calculations, due to higher resistivity of the GSST across all phases compared to GST, the electronic contribution to thermal conductivity for the crystalline phase is suppressed by more than an order of magnitude. Figure 4b shows the electronic contribution to thermal conductivity as a function of temperature at different annealing temperatures. According to this plot, the electronic contribution increases for higher annealing temperatures and reaches a maximum of 0.16 W m$^{-1}$ K$^{-1}$ at 383 °C. This means that, the electronic contribution in thermal conductivity of GSST at 383 °C is approximately 25% which reduces to 5% near room temperature.

In order to qualitatively investigate the contribution of electrons in thermal conductivity using thermal conductivity measurements, we anneal the GST and GSST at different temperatures and measure their thermal conductivity upon cooling. Since electronic contribution in GST changes significantly as a result of metal−insulator transition[48] upon heating, we expect the thermal conductivity trend as a function of temperature changes for higher annealing temperatures. For this purpose, we heat an as-deposited GST sample to temperatures close and above the phase transition, and by cooling the samples down, we record its thermal conductivity (hollow circles in Fig. 4c). The solid circles in Fig. 4c show the thermal conductivity of as-deposited GST upon heating, whereas, the hollow circles show the thermal conductivity of GST at different annealing temperatures upon cooling. According to these results, for annealing temperatures of 160 and 240 °C, the thermal conductivity of GST remains constant upon cooling to room temperature. However, when the annealing temperature raises above 300 °C, the thermal conductivity trend begins to increase with temperature (decrease upon cooling). This change in the trend of thermal conductivity as a function of temperature cannot be

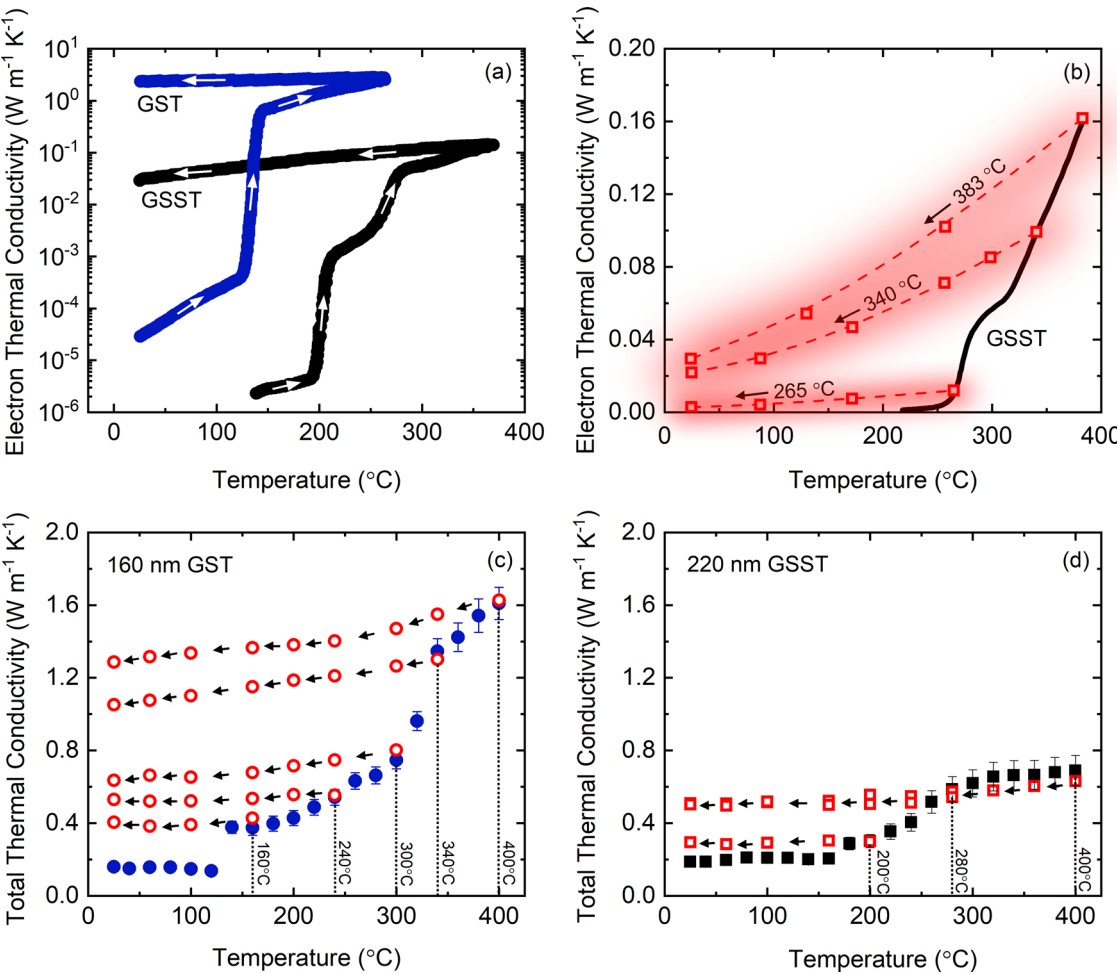

**Fig. 4 Electronic contribution to the thermal conductivity of GST and GSST. a** Electronic contribution to thermal conductivity for GST and GSST upon heating and phase transformation obtained from W−F. **b** Electron thermal conductivity of GSST as a function of temperature upon cooling at various annealing temperatures obtained from W−F. The shaded area shows 25% uncertainty associated with Lorenz number. Measured total thermal conductivity from TDTR for **c** 160 nm GST and **d** 220 nm GSST upon heating (solid symbols), and their corresponding thermal conductivity upon cooling (hollow symbols) at different annealing temperatures. The vertical dotted lines show the annealing temperatures. The uncertainty is calculated based on 10% variation in transducer thickness.

explained from a pure phononic perspective. This is because in crystalline systems, the thermal conductivity either decreases with temperature due to an increase in the anharmonic interactions or in the case of 2D or defective crystals it follows the heat capacity trend (i.e., constant above the Debye temperature). We rule out the increase in thermal conductivity due to heat capacity because we observe a constant trend up to 240 °C, indicating we are well above the Debye temperature of the GST. According to these, we can only relate the increase in thermal conductivity of GST to the enhancement in the electronic contribution. Considering the metal−insulator transition in GST near 300 °C, this is a reasonable deduction. On the other hand, repeating the same procedure for the GSST, yields a different behavior. As can be seen in Fig. 4d the thermal conductivity for the annealed cases remain constant with temperature for all annealing temperatures. Considering the high electrical resistivity of GSST compared to GST, this supports our conclusion that the thermal conductivity of GSST is reduced due to suppression of electronic contribution.

So far, using W−F law and thermal conductivity trend, we showed that the reduction in thermal conductivity of GSST could be related to the suppressed electronic contribution. Nonetheless, although W−F approximation provides a reasonable qualitative insight to the contribution of electrons in thermal conductivity,

the Lorenz number has been shown to vary with temperature on multiple occasions, and is well-known to deviate from its standard value under conditions of increased inelastic electronic collisions (e.g., "vertical processes")[50–54]. As a result, we turn to additional experimental measurements to verify this. First, we provide a direct measurement of both the electron and phonon lifetimes in the crystalline phases of GSST and GST through ultrafast mid-infrared spectroscopy[55,56].

To further investigate the varying scattering mechanisms that give rise to changes in the thermal conductivity of these chalcogenide-based PCMs, we turn to mid-infrared (MIR) pump-probe spectroscopy to directly monitor the lifetime of electronic and vibrational energy carriers in these materials. Described in more detail elsewhere[55,56], we excite the GSST and GST films with a sub-picosecond 520 nm pump pulse, thus inducing a non-equilibrium state in the film of interest. At varying time delays, a tunable-wavelength MIR probe pulse interrogates the change in reflectivity of the film due to the change in either carrier concentration or temperature[57]. At wavelengths near interband transitions (e.g., bandgap) in the films, this modulated reflectivity becomes dominated by the electronic response of the material system and is indicative of changes in the electronic density of states[56]. In contrast, for wavelengths far from such transitions, the response

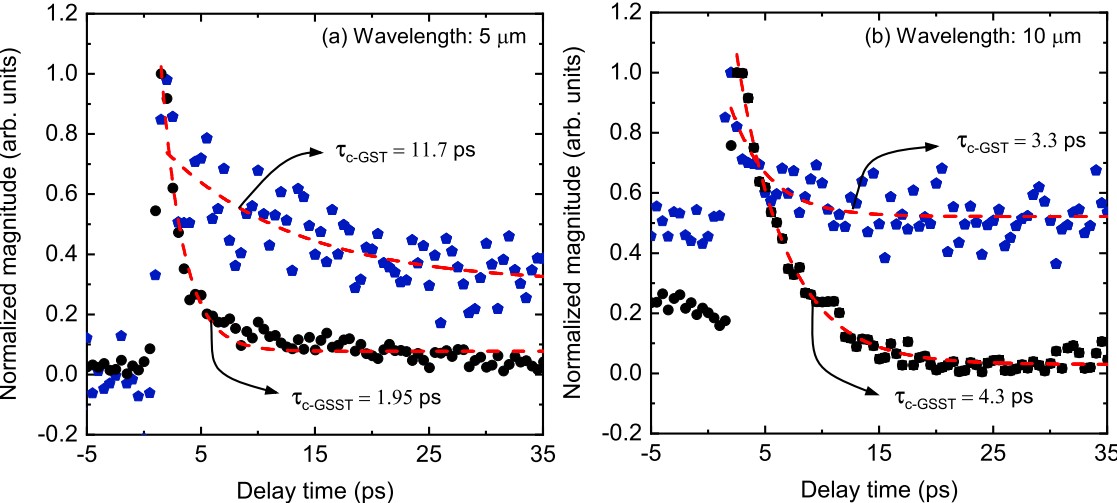

**Fig. 5 Ultrafast mid-IR pump and probe spectroscopy.** Modulated reflectivity data for our mid-infrared ultrafast spectroscopy measurements for probe wavelengths of **a** 5 μm and **b** 10 μm. The higher energy (lower wavelength) probe beam preferentially interrogates the electronic density of states upon pump excitation, demonstrating a greatly suppressed electronic lifetime of GSST compared to its GST counterpart. In contrast, the lower energy (higher wavelength) probe response is indicative of the materials' lattice response, and demonstrates similar decay times for both GST and GSST, indicating similar vibrational lifetimes for the two material systems. Note, the decay time is obtained by fitting an exponential decay to the data and the large variation in the experimental data for the GST stems from its thinner film thickness and different deposition technique.

becomes dominated by changes in the vibrational density of states, and is thus a strong indication of the change in lattice temperature[56]. The decay time of these subsystems is indicative of their relative populations; where the electron response is dictated by both the subsystem temperature and excited-state number density[58]. As such, the reported decay times between GST and GSST are indicative of a convolution of recombination processes, as well as intrinsic carrier scattering rates.

The results of this ultrafast MIR spectroscopy are shown for two select probe wavelengths in Fig. 5. For the higher energy probe beam (probe wavelength of 5 μm, Fig. 5a), we observe a greatly increased decay rate of the ultrafast signal in the c-GSST ($\tau = 1.95$ ps) film compared to the c-GST ($\tau = 11.7$ ps) counterpart. In other words, the electron decay time/scattering rate is greatly increased in GSST relative to GST films. The origin of this enhanced relaxation time can likely be attributed to inelastic electron scattering with high-frequency nuclear motion (e.g., high-frequency phonons) that are introduced from the addition of Se to the crystal basis (see Raman data in Fig. 2b); similar effects have been demonstrated, experimentally[55,59] and computationally[60–62], in other metallic systems following the incorporation of lighter atoms. While this observed suppression of the electron lifetime in GSST likely invalidates the use of L = L0 in our W−F analysis of electronic thermal conductivity, it further supports our posit that electrons in GSST do not significantly contribute to its energy transport/thermal conductivity when compared to its GST counterpart. In contrast, for probe wavelengths that dominantly interrogate the vibrational states in the films of interest, we find relatively similar scattering times, 3.3 and 4.3 ps, for both GST and GSST, respectively. In other words, the phonon lifetimes of the two material systems are comparable, which, at least qualitatively, supports the similarity in their comparable lattice contributions to thermal conductivity.

In summary, we report on the thermal properties of Ge$_2$Sb$_2$Se$_4$Te, an emerging phase change material with superior properties for photonic applications. We observe that the thermal conductivity of the GSST as a function of temperature in the amorphous phase remains constant around ~0.18 W m$^{-1}$ K$^{-1}$ up to the phase transition temperature at 180 °C. Upon phase transition and raising the temperature to 400 °C, the thermal conductivity increases to ~0.65 W m$^{-1}$ K$^{-1}$. We show that the thermal conductivity of GSST is more than a factor of 2 lower than its close cousin GST. We attribute this reduction in thermal conductivity to strong suppression of electronic contribution to the thermal conductivity of crystalline GSST. This reduction in thermal conductivity of the GSST allows for better confinement of heat near the memory cell which could potentially lead to less power consumption. However, the sluggish crystallization of GSST would require an order of magnitude longer pulse duration for crystallization that overwhelms the power consumption in these materials. Nonetheless, our work demonstrates that substituting Te with Se in GST, results in strong suppression of electronic contribution in thermal conductivity.

## Methods
**Thermal properties measurements**. The thermal conductivity, volumetric heat capacity, and sound speed reported in this study are measured using TDTR. In these measurements, the output of a Ti:sapphire laser operating at 80 MHz repetition rate is split into a pump and a probe path. Using an electro-optic modulator (EOM), the pump path is modulated at a frequency of 8.4 MHz to create an oscillatory temperature rise on the surface of the sample. As a result of this periodic temperature rise, the reflectivity of the surface changes which can be detected using a lock-in amplifier. The pump and probe beams after passing through a 10× objective are focused to Gaussian spots with sizes of 20 and 10 μm, respectively, measured using a Thorlabs beam profiler. To facilitate capturing the changes in the reflectivity, all of the samples are coated with either 80 nm of aluminum or platinum. For sound speed measurements, a different set of samples is made with no interlayer between Al and GSST or GSST and Si, Al/GSST/Si. The thickness of Al for this configuration is chosen to be 750 nm in order to avoid the appearance of multiple echoes from Al/GSST interface.

**Transmission electron microscopy**. Cross-sections samples for TEM were made using a Thermo Fisher Helios Dual-Beam Focused Ion Beam. Over 1 μm of Pt-protective coating was deposited with the electron beam prior to ion exposure to minimize radiation damage. Milling was initially performed with 30 kV Ga-ions the sequentially decreasing current to a thickness of ~750 nm. Then, the Ga-ion energy was reduced to 8 kV for further thinning and a final cleaning was performed at 5 kV. TEM is from a Thermo Fisher Themis Z-STEM operating at 200 kV. The selected area aperture for selected area electron diffraction patterns is placed partially in the Si substrate for accurate calibration of reciprocal-space. For the 20 nm thin-films, the Si filled the majority of the selected-area aperture, due to the minimum aperture size being >> 20 nm, and therefore the Si scattering to Bragg diffraction peaks was much higher intensity than the film's scattering. Energy dispersive X-ray spectroscopy was performed on the Themis Z-STEM equipped

with a Super X EDS system (4 silicon drift detectors) in scanning mode with a 30 mrad convergence and probe current of 400 pA.

**Reporting summary**. Further information on research design is available in the Nature Research Reporting Summary linked to this article.

## Data availability
The data that support the findings of this study are available from the corresponding author upon reasonable request.

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

## Acknowledgements

We appreciate the support from Western Digital Technologies, Inc. This manuscript is based upon work supported by Office of Naval Research under Award No. N00014-20-1-2686 and National Science Foundation Award No. 162601. Y.Z., C.R., and J.H. acknowledge Dr. Kathleen Richardson for providing the GSST source materials and funding support provided by the Defense Advanced Research Projects Agency Defense Sciences Office (DSO) Program: EXTREME Optics and Imaging (EXTREME) under Agreement No. HR00111720029. Utilization of the Thermo Fisher Themis Z-STEM and Thermo Fisher Helios Dual-Beam Focused Ion Beam instruments within UVa's Nanoscale Materials Characterization Facility (NMCF) was fundamental to this work.

## Author contributions

K.A., J.A.T., and P.E.H. designed the experiment. Y.Z., C.R., J.N., J.C.R., and J.H. made the samples. K.A., J.A.T., M.S.B.H., and D.H.O. performed the experiments. E.R.H. performed the TEM characterization. K.A., J.A.T., and P.E.H. wrote the manuscript.

## Competing interests

The authors declare no competing interests.
