## [Peer Review File · Nature Communications]

Suppressed electronic contribution in thermal conductivity of Ge₂Sb₂Se₄TeREVIEWER COMMENTS

Reviewer #1 (Remarks to the Author):

In this manuscript, the authors present a suite of measurements on the thermal properties of the phase change material GSST ($\text{Ge}_{2}\text{Sb}_{2}\text{Se}_{4}\text{Te}$), which include TEM, TDTR, sound speed measurements, and an interesting and unique comparison of these to ultrafast mid IR spectroscopy. The main result is the demonstration of a much reduced thermal conductivity in these films compared to the related parent compound with no Se substitution (GST) that the authors have previously investigated. Their work rather elegantly indicates that the reduction is related to strongly enhanced electron scattering in the Se-doped material. The coupling of the spectroscopy with the thermal measurements allows this result, though the authors do not really delve into any possible physical mechanism for this electron scattering. Nevertheless, this is an important, well-executed, and (mostly) clearly described experiment that I suspect will make a nice contribution to Nature Communications. I would like the authors to address a few questions, concerns, and minor editorial issues. I list these below.

- 1). In the first paragraph, the authors motivate their study by references to "biologically-inspired" neuromorphic computing. They further state that one of the promises of this type of computing is "higher bit density." I find this a bit confusing, since the typical size of neurons is enormous compared to current CMOS technology. I would suggest they look at this sentence more carefully and try to be more specific about which potential advantage of neuromorphic computers they mean here. Higher connectivity or faster computation seem more accurate than high bit density.
- 2). In the second paragraph of the right-hand column on the first page, the authors introduce a figure of merit for photonic phase change materials. They do not explain the Δn or Δk in this ratio. One could guess this, but for a journal with broad readership such as Nat. Comms., please define these symbols.
- 3). The presentation of some structural characterization on these films is welcome, but the authors do not attempt much quantitative analysis of these diffraction patterns. It would be interesting to include, for example, the average real-space distance that corresponds to the diffuse scattering ring they identify in the amorphous material. Does this distance correspond to something meaningful in the material? I am not sure it is possible based on the data in-hand, but some consideration of the in-plane and out-of-plane average grain size in the crystalline films might be useful when considering potential sources of the inelastic electron scattering that they identify. A comparison of these grain sizes for the GST and GSST films could be important.
- 4). Can the authors label or otherwise comment on the large microraman peak at lower wavenumber that lies below both the crystalline and amorphous peak in Fig. 2b? Again, for the broad readership, explaining this can make things a bit more accessible.
- 5). The caption of Fig. 2d says the amorphous data is shown in red diamonds, but the diamonds in the actual figure are black.
- 6). The labels in Fig 1. suggests that the film capping the GSST is aluminum (metal) but the description given when discussing the thermal conductivity measurements says that alumina (Al_2O_3) films were used. Can the authors please clarify this issue?
- 7). The authors currently only present the expected annealing and temperature dependence of the electronic contribution to k (calculated from the WF law) for the GSST (in Fig. 4b). I would suggest adding a similar plot for the GST. I suspect that the trends in that plot will follow their measured data in Fig. 4c at least qualitatively quite well.

8). As I mentioned earlier, there is currently little suggestion or consideration of a physical mechanism that could cause the inelastic electron scattering that the authors have observed in the GSST. It may be appropriate for the authors to consider some possible physical mechanisms. For example, the defect scattering that the authors do mention is typically quite elastic. Would there be some reason that scattering off the Se sites could cost the electron some energy? Could some resonant inelastic scattering occur due to a sort of rattling mode, if the Se is less tightly bound in the structure, as is commonly discussed in the skutterudites, for example?

Reviewer #2 (Remarks to the Author):

This study investigates the thermal properties of GSST film by implementing an ultrafast mid-infrared pump-probe spectroscopy technique. Upon substituting Te with Se, the thermal conductivity of GSST film is greatly decreased. The reason is found to be the drastic change in electronic lifetimes of GSST. I recommend this work to be published after addressing the following issues.

1. What is the reason for the incomplete crystallization of GSST film? Did the authors perform the XRD analysis? In addition, did the authors observe secondary phases?
2. In Fig. 2d, the sound speed increases with increasing the thickness. Can the authors comment the reason?
3. In principle, the heat capacity is only determined by the material's chemical composition. What is the reason for the different heat capacity of the amorphous and annealed GSST films?
4. Why did the authors used the Lorenz number $2.44 \times 10^{-8} \text{ W W K}^{-2}$ to calculate the electronic contribution in thermal conductivity?
5. Alloying Se in GST should introduce additional point defect scattering to the phonons, but its role on thermal conductivity reduction is not discussed in this work.
6. Can the authors comment why the electron scattering time is longer than the vibrational scattering time?
7. The measured k for the amorphous GSST film is already lower than the $k_{\text{min,p}}$. How to understand this phenomenon?

Reviewer #1 (Remarks to the Author):

In this manuscript, the authors present a suite of measurements on the thermal properties of the phase change material GSST ($\text{Ge}_2\text{Sb}_2\text{Se}_4\text{Te}$), which include TEM, TDTR, sound speed measurements, and an interesting and unique comparison of these to ultrafast mid IR spectroscopy. The main result is the demonstration of a much reduced thermal conductivity in these films compared to the related parent compound with no Se substitution (GST) that the authors have previously investigated. Their work rather elegantly indicates that the reduction is related to strongly enhanced electron scattering in the Se-doped material. The coupling of the spectroscopy with the thermal measurements allows this result, though the authors do not really delve into any possible physical mechanism for this electron scattering. Nevertheless, this is an important, well-executed, and (mostly) clearly described experiment that I suspect will make a nice contribution to Nature Communications. I would like the authors to address a few questions, concerns, and minor editorial issues. I list these below.

1). In the first paragraph, the authors motivate their study by references to "biologically-inspired" neuromorphic computing. They further state that one of the promises of this type of computing is "higher bit density." I find this a bit confusing, since the typical size of neurons is enormous compared to current CMOS technology. I would suggest they look at this sentence more carefully and try to be more specific about which potential advantage of neuromorphic computers they mean here. Higher connectivity or faster computation seem more accurate than high bit density.

Authors Response:

We appreciate the reviewer's response and suggestions. To address the reviewer's concern regarding the discontinuity in the motivation section of the introduction, we modified the following statement in the introduction, Pg. 01, 1st paragraph:

"Biologically-inspired neuromorphic computing is one of the more promising alternatives to transistor-based technologies that not only offers significantly higher degree of connectivity between the memory nodes leading to faster computation and less power consumption, but also allows for simultaneous storage and processing of information within the memory cell^{7,8}."

2). In the second paragraph of the right-hand column on the first page, the authors introduce a figure of merit for photonic phase change materials. They do not explain the Δn or Δk in this ratio. One could guess this, but for a journal with broad readership such as Nat. Comms., please define these symbols.

Authors Response:

We appreciate the reviewer's comment regarding the ambiguity of figure of merit. To address the reviewer's concern, we modified the following statement in the introduction, Pg. 02, 1st paragraph:

"Recently, a new class of phase change materials, $\text{Ge}_2\text{Sb}_2\text{Se}_4\text{Te}$ (GSST), has emerged which offers superior properties in regards to photonic applications, such as broadband transparency for the wavelengths in the range of 1–18.5 μm , significant contrast in the refractive index (Δn) between the phases with low optical loss (Δk) leading to a large figure of merit ($\Delta n/\Delta k$), and improved thermal stability²²⁻²⁴."

3). The presentation of some structural characterization on these films is welcome, but the authors do not attempt much quantitative analysis of these diffraction patterns. It would be interesting to include, for example, the average real-space distance that corresponds to the diffuse scattering ring they identify in the amorphous material. Does this distance correspond to something meaningful in the material? I am not sure it is possible based on the data in-hand, but some consideration of the in-plane and out-of-plane average grain size in the crystalline films might be useful when considering potential sources of the inelastic electron

scattering that they identify. A comparison of these grain sizes for the GST and GSST films could be important.

Authors Response: The authors appreciate the Reviewer's advice to attain quantitative rather than qualitative data from the (S)TEM characterization. The intent of the (S)TEM characterization is to prove three points: 1) confirm the phase of the films in amorphous and crystalline states, 2) investigate the homogeneity of the film, and 3) determine an accurate thickness of the thin-films. In proving these points we ensure the accuracy of our thermal characterization and the existence of the phase transformation.

To the reviewers point, quantitative analysis is always best when possible. Unfortunately, with the presented diffraction patterns a quantitative analysis of real-space distances would not be entirely accurate. In the case of crystalline materials, one can obtain interplanar spacings by taking the reciprocal of a Bragg diffraction peak's scattering vector. This is because of the long-range periodic real-space order of the lattice, which is also represented in reciprocal-space. For amorphous materials, which lack long-range order, the contribution of all scattering (Fourier) components need to be considered. This is where experiments calculating electron-pair-distribution functions has revolutionized the study of amorphous solids and metallic glasses [1,2]. In our experiments, the standard techniques used for an electron-pair-distribution function analysis of the amorphous scattering rings would be overwhelmed by the high-intensity Si Bragg peaks that are also present in the data.

As to the question regarding grain size, the authors have adjusted the image contrast to provide a better visual representation. The diffraction contrast typically used to analyze grain size is weak in the images. None the less, there are a few grains that have vaguely discernible boundaries in the 150 nm film. We have quantified the range of grain size as 20-50 (in-plane) * 10-110 nm (out-of-plane) which agrees with our previous published TEM analysis [3]. On Pg. 02 of the manuscript, we have included the following sentences and updated figure to further describe the microstructure:

“The bright-field transmission electron microscope (TEM) image (Fig. 1(g)) of the 150 nm film shows grains with sizes ranging from 20-50 (in-plane) x 10-110 (out-of-plane). The grains in the 20 nm film were equiaxed and span the thickness of the film.”

References

[1] Im, Soohyun, et al. "Direct determination of structural heterogeneity in metallic glasses using four-dimensional scanning transmission electron microscopy." *Ultramicroscopy* 195 (2018): 189-193.

[2] Voyles, Paul M., and John R. Abelson. "Medium-range order in amorphous silicon measured by fluctuation electron microscopy." *Solar energy materials and solar cells* 78.1-4 (2003): 85-113.

[3] Zhang, Yifei, et al. "Broadband transparent optical phase change materials for high-performance nonvolatile photonics." *Nature communications* 10.1 (2019): 1-9.

4). Can the authors label or otherwise comment on the large microraman peak at lower wavenumber that lies below both the crystalline and amorphous peak in Fig. 2b? Again, for the broad readership, explaining this can make things a bit more accessible.

Authors Response: We thank the reviewer for raising this point. While it might look like another peak, the signal at lower wavenumber corresponds to the baseline signal with a sharp cut at 50 cm^{-1} due to the system's lower bound. Large signals are typical for small Raman shifts, given that the wavenumbers are closer to that of the pump and higher noise is measured. While we have removed the overall baseline, the fitting was not good enough to reduce the signal at lower shifts without loss of information. In many other experiments, and given the difference between both states, we tend to completely avoid the baseline removal if the unequivocal signal for both amorphous and crystalline peaks are clearly distinguished (see Ref [1]).

To avoid confusing it with a peak, we have edited the figure such that the shortest Raman shift value matches our system's lower bound:

Reference:

[1] Ríos, Carlos, et al. "Multi-level electro-thermal switching of optical phase-change materials using graphene." *Advanced Photonics Research* 2.1 (2021): 2000034.

5). The caption of Fig. 2d says the amorphous data is shown in red diamonds, but the diamonds in the actual figure are black.

Authors Response:

We thank the reviewer for pointing out to this mislabeling. We corrected this in the revised version of the manuscript.

6). The labels in Fig 1. suggests that the film capping the GSST is aluminum (metal) but the description

given when discussing the thermal conductivity measurements says that alumina (Al₂O₃) films were used. Can the authors please clarify this issue?

Authors Response: We appreciate the reviewer's comment regarding the discrepancy in the labeling of the Fig. 1 and the text in the manuscript. To clarify this, we need to mention that we used multiple sample batches for various measurements. The samples used for TEM are those that are used for sound speed measurements where there is no interlayer between the Al and GSST. However, for the thermal conductivity measurements, we used a different set of samples where the structure is Pt/Al₂O₃/GSST/Si. The use of Pt as the transducer enables us to go to higher temperatures for thermal conductivity measurements compared to the Al and the addition of Al₂O₃ (7 nm) is to prevent any possible reaction between Pt and GSST.

7). The authors currently only present the expected annealing and temperature dependence of the electronic contribution to k (calculated from the WF law) for the GSST (in Fig. 4b). I would suggest adding a similar plot for the GST. I suspect that the trends in that plot will follow their measured data in Fig. 4c at least qualitatively quite well.

Authors Response: We would like to thank the reviewer for the suggestion. Although adding a similar electronic contribution to the thermal conductivity of GST at different annealing temperatures would be great, unfortunately, we do not have these data available. However, in order to address reviewer's concern and demonstrate our hypothesis regarding the impact of electron in thermal conductivity of GST in crystalline phase, we turn to existing data available in literature. Lee et al. [1] have measured the electrical resistivity of GST at different annealing temperatures and after calculating the electronic contribution from their electrical resistivity data, we observe that the electronic contribution in GST is significantly increased at higher annealing temperatures. Although their electrical resistivity for 400 °C annealed case suggests that the k_e is around ~3 W/m/K, we need to consider that this value is a rough W-F estimation and does not necessarily mean the electronic contribution should be as high. A slight error in the electrical resistivity could change the electronic contribution significantly. As a result, these data are only useful for comparison purposes and the absolute value should not be considered.

Looking at the available data for the electrical resistivity of the h-GST amongst heavily cited studies, we observe a large difference between the reported values. The table below shows the

measured resistivity reported by various groups and the estimated thermal conductivity using W-F equation.

	Resistivity for h-GST (mΩ cm)	Temperature (K)	Ge ₂ Sb ₂ Te ₅ Thermal Conductivity* (W/mK)
Kato & Tanaka ^[2]	~3	400	0.244
Burr & coworkers ^[3]	~2	300	0.366
Wuttig & coworkers ^[4]	~0.8	400	0.915
Cahill & coworkers ^[5]	~0.585	300	1.251
Calarco & coworkers ^[6]	~0.320	275	2.287
Lee & coworkers ^[1]	~0.28	300	2.604

* Thermal conductivity due to electron contribution

This significant difference between the electrical resistivities in different studies could be partly due to the different deposition process, composition variation, annealing time, or different measurement techniques. Due to existence of these variations between different studies, we refrained from using their results in our study in order to avoid miscommunications.

Reference

- [1] Lee, Jaeho, et al. "Phase purity and the thermoelectric properties of Ge₂Sb₂Te₅ films down to 25 nm thickness." *Journal of Applied Physics* 112.1 (2012): 014902.
- [2] Kato, Takayuki, and Keiji Tanaka. "Electronic properties of amorphous and crystalline Ge₂Sb₂Te₅ films." *Japanese journal of applied physics* 44.10R (2005): 7340.
- [3] Nirschl, T., et al. "Write strategies for 2 and 4-bit multi-level phase-change memory." 2007 IEEE International Electron Devices Meeting. IEEE, 2007.
- [4] Siegrist, T., et al. "Disorder-induced localization in crystalline phase-change materials." *Nature materials* 10.3 (2011): 202-208.
- [5] Lyee, Ho-Ki, et al. "Thermal conductivity of phase-change material Ge₂Sb₂Te₅." *Applied Physics Letters* 89.15 (2006): 151904.
- [6] Bragaglia, Valeria, et al. "Metal-insulator transition driven by vacancy ordering in GeSbTe phase change materials." *Scientific reports* 6.1 (2016): 1-7.

8). As I mentioned earlier, there is currently little suggestion or consideration of a physical mechanism that could cause the inelastic electron scattering that the authors have observed in the GSST. It may be appropriate for the authors to consider some possible physical mechanisms. For example, the defect scattering that the authors do mention is typically quite elastic. Would there be some reason that scattering off the Se sites could cost the electron some energy? Could some resonant inelastic scattering occur due to a sort of rattling mode, if the Se is less tightly bound in the structure, as is commonly discussed in the skutterudites, for example?

Authors Response: We appreciate the reviewer's comments and suggestions regarding the mechanisms for the electronic suppression in GSST relative to its GST counterpart. We agree that the referenced defect/mass scattering is an intrinsically *elastic* scattering process. However, this is under the assumption that the additional Se atoms are 'impurities' or defects in an otherwise-crystalline material. In reality, the addition of Se to the crystal's basis affects both the electronic and phononic density of states, and thus the electron-phonon interactions of the material. Such

electron-phonon scattering processes are inelastic and reduce the energy a given electron. Specifically, as Se is lighter mass than both Sb and Te, GSST will have intrinsically higher frequency vibrations than that of GST. The nuclear motion associated with such high frequency vibrations has been experimentally and computationally demonstrated to enhance electron-phonon scattering rates and lead to reduced electronic lifetimes in other material systems. The specific modes generated by this additional light atom could certainly be akin to that of a ‘rattling mode’ discussed in skutterudites, but the calculations of the exact frequencies of the new vibrational modes generated through the addition of Se are perhaps outside the scope of this work due to the computational rigor required.

To incorporate these specific details and increase clarity to potential readers, we have included additional text to our revised manuscript. This revised text can be found on Pg. 6 and now reads as:

“...In other words, the electron decay time/scattering rate is greatly increased in GSST relative to GST films. The origin of this enhanced relaxation time can likely be attributed to inelastic electron scattering with high frequency nuclear motion (e.g., high frequency phonons) that are introduced from the addition of Se to the crystal basis (see Raman data in Fig. 2b); similar effects have been demonstrated, experimentally [1,2] and computationally [3-5], in other metallic systems following the incorporation of lighter atoms.”

References

- [1] Olson, David H., et al. "Band alignment and defects influence the electron–phonon heat transport mechanisms across metal interfaces." *Applied Physics Letters* 118.16 (2021): 163503.
- [2] Tomko, John A., et al. "Long-lived modulation of plasmonic absorption by ballistic thermal injection." *Nature nanotechnology* 16.1 (2021): 47-51.
- [3] Lu, Teng-Fei, et al. "Control of charge carrier dynamics in plasmonic Au films by TiO_x substrate stoichiometry." *The journal of physical chemistry letters* 11.4 (2020): 1419-1427.
- [4] Wang, Yi-Siang, et al. "Electron–Phonon Relaxation at Au/Ti Interfaces Is Robust to Alloying: Ab Initio Nonadiabatic Molecular Dynamics." *The Journal of Physical Chemistry C* 123.37 (2019): 22842-22850.
- [5] Zhou, Xin, et al. "Thin Ti adhesion layer breaks bottleneck to hot hole relaxation in Au films." *The Journal of chemical physics* 150.18 (2019): 184701.

Reviewer #2 (Remarks to the Author):

This study investigates the thermal properties of GSST film by implementing an ultrafast mid-infrared pump-probe spectroscopy technique. Upon substituting Te with Se, the thermal conductivity of GSST film is greatly decreased. The reason is found to be the drastic change in electronic lifetimes of GSST. I recommend this work to be published after addressing the following issues.

1. What is the reason for the incomplete crystallization of GSST film? Did the authors perform the XRD analysis? In addition, did the authors observe secondary phases?

Authors Response: We appreciate the reviewer's remark about the incomplete crystallization observed in the diffraction pattern. To address reviewer's concern, we would like to point out that the incomplete crystallization observed in the TEM images is probably due to damage caused during the FIB process. It is not uncommon for tellurium- and selenium-based alloys to get damaged during post-processing as they have low melting temperatures (450°C and 220°C, respectively). In our previous work on GST [1], we performed in-situ heating TEM and we observed that a complete crystallization occurs upon phase transformation at ~150°C. The GSST films are highly prone to irradiation damage from the ion-beam used during the FIB TEM sample preparation and the electron beam during TEM imaging. As a result, the appearance of amorphous ring in the diffraction pattern may not be from incomplete recrystallization but could be due to the Ga-ions damaging the sample.

In addition, the authors did not see any evidence of chemically distinguishable secondary phases. We have provided a new supporting figure showing STEM-EDS analysis of the 220 nm sample. The following has been added to the supporting information:

“Figure S2. STEM-EDS spectrum image with (a) diffraction contrast from an annular darkfield detector showing multiple grains with uniform (b) Ge, (c) Se, (d) Sb, and (e) Te compositions. The

Si and Al layers are shown in (f) along with thin oxides present at each interface. The composite image of (a-e) is shown in (g), where a line profile is indicated. The quantitative compositions along the line profile are shown in (h)."

"STEM-EDS was performed using a 400 pA beam current and the results are shown in Figure S. The annular darkfield image in Figure S(a) shows multiple equiaxed and non-equiaxed grains. The composition of the various grains do not have differing compositions in anyway correlated to the morphologies in the darkfield signal, as shown in Figure S(b-e). A compositional gradient is present across the thickness of the films, as shown in Figure S(h). The gradient may in part be from x-ray absorption affects that are thickness dependent or could be a result of growth conditions. It is striking that such a large gradient is present, and the gradient may explain the preferential non-equiaxed grain morphology being present at the bottom of the film and not the top."

References

[1] Aryana, Kiumars, et al. "Interface controlled thermal resistances of ultra-thin chalcogenide-based phase change memory devices." *Nature communications* 12.1 (2021): 1-11.

2. In Fig. 2d, the sound speed increases with increasing the thickness. Can the authors comment the reason?

Authors Response: This is a fair question that perhaps deserves more attention. A similar observation has been shown in other crystalline material systems [1-4], but is rarely noted in their amorphous counterparts. Although we observe a similar reduction in the sound speed of crystalline GST with thickness, we have not been able to conclusively identify the reasons for this. The sound speed primarily depends on the density and the elastic modulus of materials. However, for crystalline thin films, effects akin to substrate clamping or lattice matching can occur, which alter the local density and thus mechanical properties of the atomic layers near the film/substrate interface. This change would be most prominent in crystalline films, where the periodic lattice would be perturbed near the interface; in contrast, amorphous films are already lacking such periodicity, and the average properties would go relatively unchanged. Thus, for very thin films that approach the length of this atomic perturbation, one would expect the most drastic change in atomic number density and mechanical properties of the film.

While we suspect the role of substrate-clamping to be the most likely cause of the observed changes in sound velocity for crystalline GSST films, the necessary experiments to verify this are outside the scope of this work.

References

[1] Ma, Weigang, et al. "Comprehensive study of thermal transport and coherent acoustic-phonon wave propagation in thin metal film–substrate by applying picosecond laser pump–probe method." *The Journal of Physical Chemistry C* 119.9 (2015): 5152-5159.

[2] Wright, O. B. "Thickness and sound velocity measurement in thin transparent films with laser picosecond acoustics." *Journal of Applied Physics* 71.4 (1992): 1617-1629.

[3] Hostetler, J. L., A. N. Smith, and P. M. Norris. "Thin-film thermal conductivity and thickness measurements using picosecond ultrasonics." *Microscale Thermophysical Engineering* 1.3 (1997): 237-244.

[4] Eesley, Gary L., Bruce M. Clemens, and Carolyn A. Paddock. "Generation and detection of picosecond acoustic pulses in thin metal films." *Applied physics letters* 50.12 (1987): 717-719.

3. In principle, the heat capacity is only determined by the material's chemical composition. What is the reason for the different heat capacity of the amorphous and annealed GSST films?

Authors Response: This is a great question. The short answer is that the heat capacity of the film changes due to densification of the GSST upon crystallization. Our reported value is the *volumetric* heat capacity which is the specific heat (J/g/K) * density (g/cm³). The reviewer is correct that the specific heat is not supposed to change between the phases. However, volumetric heat capacity largely depends on the atomic mass as well as density of the material.

According to our TEM measurements, there is a ~5% reduction in the thickness upon amorphous to crystalline phase transformation which agrees with previous observations [1,2]. Now, there are two scenarios that could take place here; first, it is possible that due to strain effects from the substrate and the transducer layer, the film density does not change in the in-plane direction (i.e. only the thickness of the film changes). This means that the change in film thickness (~5%) is comparatively equal to changes in the density upon crystallization. This scenario is consistent with previous measurements of density in GST films where the changes in thickness of the film is similar to the changes in the density [1].

However, if the 5% densification in the film occurs in all directions, xyz, then the change in the density is larger than change in the thickness. Assuming isotropic change in all direction (~5%), we estimate the change in the film density ($d_{\text{amorphous}}^3/d_{\text{crystalline}}^3$) to be ~15%.

Now, according to our heat capacity measurements, the heat capacity changes by ~20% upon crystallization. Considering the uncertainty associated with our measurements this means that the heat capacity can change from minimum of 5% to maximum of 30%, which is within the range of both scenarios.

References

[1] Njoroge, Walter K., Han-Willem Wöltgens, and Matthias Wuttig. "Density changes upon crystallization of Ge₂Sb₂Te₄ films." *Journal of Vacuum Science & Technology A: Vacuum, Surfaces, and Films* 20.1 (2002): 230-233.

[2] Aryana, Kiumars, et al. "Interface controlled thermal resistances of ultra-thin chalcogenide-based phase change memory devices." *Nature communications* 12.1 (2021): 1-11.

4. Why did the authors used the Lorenz number $2.44 \times 10^{-8} \text{ W } \Omega \text{ K}^{-2}$ to calculate the electronic contribution in thermal conductivity?

Authors Response: We would like to thank the reviewer for raising this important question regarding our WF calculation. According to Thesberg et al. [1] the Lorenz number for semiconductor is expected to be in the range of 1.49 for nondegenerate semiconductors to $2.45 \times 10^{-8} \text{ W } \Omega \text{ K}^{-2}$ for degenerate semiconductors and metals. In this regard, since GST is a degenerate semiconductor in its hexagonal phase [2] we chose the $2.44 \times 10^{-8} \text{ W } \Omega \text{ K}^{-2}$ for our thermal conductivity calculations. In addition, since the electrons contribution to thermal conductivity in the GSST is so low, even if the Lorenz number is changed by a factor of two, this would only lead to a thermal conductivity of 0.06 W/m/K which is around 11% of total thermal conductivity in GSST.

In order to address the reviewer's concern, we add error bars (shaded area) to the electron thermal conductivity plot using 25% uncertainty in the Lorenz number:

References

[1] Thesberg, Mischa, Hans Kosina, and Neophytos Neophytou. "On the Lorenz number of multiband materials." *Physical Review B* 95.12 (2017): 125206.

[2] Kato, Takayuki, and Keiji Tanaka. "Electronic properties of amorphous and crystalline Ge₂Sb₂Te₅ films." *Japanese journal of applied physics* 44.10R (2005): 7340.

5. Alloying Se in GST should introduce additional point defect scattering to the phonons, but its role on thermal conductivity reduction is not discussed in this work.

Authors Response: We would like to thank the reviewer for pointing out to this important phonon scattering mechanism. Although we have briefly discussed this in the original draft of the paper, we rephrase our discussion with more details to highlight the discussion regarding the alloy scattering as follows:

"In order to investigate how thermal properties GST change upon Se substitution from a pure vibrational mode perspective, disregarding the contribution from electrons, both Se and Te are 2-coordinated elements, and therefore, Se substitutions should not alter the degree of connectivity and bonding network in the system, leaving minimal impact on the thermal conductivity from a topological perspective⁴². However, the insertion of Se as an additional element is expected to increase the scattering rate. At the same time, Se is a lighter element compared to Te and should lead to an increase in thermal conductivity. As a result of these competing events and to gain a deeper understanding of the dominant phonon scattering mechanisms in GSST, we turn to analytical models that provide an estimation for the thermal conductivity based on kinetic theory of gases, generally referred to as the minimum limit to thermal conductivity⁴³. According to Cahill and Pohl⁴⁴, the minimum scattering length of vibrational carriers, in this case phonons, is on the order of one half of their wavelength:"

6. Can the authors comment why the electron scattering time is longer than the vibrational scattering time?

Authors Response:

This is an excellent question from the reviewer. In our infrared pump-probe experiments, we measure the relative lifetime of electrons and phonons by monitoring the wavelength-dependent response of both GST and GSST. This is done by measuring either the change in distribution of the electron or phonon subsystem independently. In materials that obey the semi-classical two-temperature model (e.g., free electron metals), the electron and lattice responses are indicative of their independent temperatures. In other words, due to rapid electron thermalization times, the electron and phonon subsystems are well-described by their equilibrium distribution functions

(Fermi-Dirac and Bose-Einstein distributions, respectively). This allows one to gain insight to directly measure the time of electron-phonon scattering processes.

However, in GST/GSST and other semiconductors, the ultrafast response due to pulsed excitation is more complicated – the electron distribution is a convolution of both the electron subsystem’s temperature *and* the excited state number density. This additional contribution from number density decays following the timescale of exciton recombination and/or electron diffusion. Critically, the rate of this additional contribution to the transient response can be longer than that of the fundamental vibrational scattering times of the material system. However, in our current work, we are merely exhibiting the *difference* in electron decay rates between GST and GSST – quantifying the absolute value of the electron scattering rates requires a significantly more in-depth description of the joint density of states that is outside the scope of this work.

To increase clarity to potential readers, we have added additional text to Page 06 of our revised manuscript to address this critical aspect of our work. The revised text now reads as:

“At wavelengths near interband transitions (e.g., band gap) in the films, this modulated reflectivity becomes dominated by the electronic response of the material system and is indicative of changes in the electronic density of states. In contrast, for wavelengths far from such transitions, the response becomes dominated by changes in the vibrational density of states, and is thus a strong indication of the change in lattice temperature⁵⁴. The decay time of these subsystems is indicative of their relative populations; where the electron response is dictated by both the subsystem temperature and excited-state number density⁵⁶. As such, the reported decay times between GST and GSST are indicative of a convolution of recombination processes as well as intrinsic carrier scattering rates.”

7. The measured k for the amorphous GSST film is already lower than the $\kappa_{\min,p}$. How to understand this phenomenon?

Authors Response: We appreciate the reviewer’s comment about the lower thermal conductivity measured for the amorphous GSST than the predicted minimum limit. To address this, we need to mention that the reported $\kappa_{\min,p}$ in the paper is for the crystalline phase of GSST. The $\kappa_{\min,p}$ for the amorphous GSST is ~ 0.30 W/m/K which is still higher than the measured value. This has been observed in prior works as well [1]. The primary reason for this discrepancy stems from the fact that the $\kappa_{\min,p}$ provides a reasonable approximation for systems where the *phonon* or *propagons* are the primary heat carriers. In low thermal conductivity materials such as amorphous alloys, the heat transfer mechanism is better described by non-propagating, delocalized atomic vibration referred to as *diffusons* and the minimum limit for the thermal conductivity is estimated from the diffuson-mediate thermal conductivity [2]:

$$\kappa_{\min,D} = 0.76 k_B n^{2/3} v_g$$

where k_B is the Boltzmann constant, v_g is the average sound velocity of the material, and n is the number density. Using this formalism, the thermal conductivity of amorphous GSST is estimated to be ~ 0.20 W/m/K which agrees well with the measured value from TDTR.

References

- [1] Lyeo, Ho-Ki, et al. "Thermal conductivity of phase-change material Ge₂Sb₂Te₅." Applied Physics Letters 89.15 (2006): 151904.
- [2] Agne, Matthias T., Riley Hanus, and G. Jeffrey Snyder. "Minimum thermal conductivity in the context of diffuson-mediated thermal transport." Energy & Environmental Science 11.3 (2018): 609-616.

REVIEWERS' COMMENTS

Reviewer #1 (Remarks to the Author):

The authors have provided a revised manuscript with thoughtful and complete responses to my questions, and changes to the manuscript where appropriate. These changes clarify some issues with the original submission. I am now happy to recommend publication of this manuscript in Nature Communications.

Reviewer #2 (Remarks to the Author):

The authors have satisfactorily answered all the comments of the reviewer. I recommend it for publication without further review.